# Analysis of Survival and Response to Lenvatinib in Unresectable Hepatocellular Carcinoma

**DOI:** 10.3390/cancers14020320

**Published:** 2022-01-10

**Authors:** Kei Amioka, Tomokazu Kawaoka, Masanari Kosaka, Yusuke Johira, Yuki Shirane, Ryoichi Miura, Serami Murakami, Shigeki Yano, Kensuke Naruto, Yuwa Ando, Yumi Kosaka, Yasutoshi Fujii, Kenichiro Kodama, Shinsuke Uchikawa, Hatsue Fujino, Atsushi Ono, Takashi Nakahara, Eisuke Murakami, Wataru Okamoto, Masami Yamauchi, Michio Imamura, Nami Mori, Shintaro Takaki, Keiji Tsuji, Keiichi Masaki, Yoji Honda, Hirotaka Kouno, Hiroshi Kohno, Takashi Moriya, Noriaki Naeshiro, Michihiro Nonaka, Hideyuki Hyogo, Yasuyuki Aisaka, Takahiro Azakami, Akira Hiramatsu, Hiroshi Aikata

**Affiliations:** 1Department of Gastroenterology and Metabolism, Graduate School of Biomedical and Health Sciences, Hiroshima University, Hiroshima 734-8551, Japan; amioka@hiroshima-u.ac.jp (K.A.); kawaokatomo@hiroshima-u.ac.jp (T.K.); m0607k@hiroshima-u.ac.jp (M.K.); jyusuke9@hiroshima-u.ac.jp (Y.J.); yuki0415@hiroshima-u.ac.jp (Y.S.); ryoichim@hiroshima-u.ac.jp (R.M.); serami@hiroshima-u.ac.jp (S.M.); yano0319@hiroshima-u.ac.jp (S.Y.); uzumaki1@hiroshima-u.ac.jp (K.N.); yuwando@hiroshima-u.ac.jp (Y.A.); eryu-f-airy1225@hiroshima-u.ac.jp (Y.K.); fujiiyasu@hiroshima-u.ac.jp (Y.F.); kodama@hiroshima-u.ac.jp (K.K.); shinuchi@hiroshima-u.ac.jp (S.U.); fujino920@hiroshima-u.ac.jp (H.F.); atsushi-o@hiroshima-u.ac.jp (A.O.); nakahara@hiroshima-u.ac.jp (T.N.); emusuke@hiroshima-u.ac.jp (E.M.); myamauchi@hiroshima-u.ac.jp (M.Y.); mimamura@hiroshima-u.ac.jp (M.I.); 2Department of Clinical Oncology, Graduate School of Biomedical and Health Sciences, Hiroshima University, Hiroshima 734-8551, Japan; wokamoto@hiroshima-u.ac.jp; 3Department of Gastroenterology, Hiroshima Red Cross Hospital & Atomic-bomb Survivors Hospital, Hiroshima 730-8619, Japan; nami7373@star.ocn.ne.jp (N.M.); shintakaki@hiroshima-med.jrc.or.jp (S.T.); k-tsuji@hiroshima-med.jrc.or.jp (K.T.); 4Department of Gastroenterology, Hiroshima City Asa Citizens Hospital, Hiroshima 731-0293, Japan; masakik8097@hiroshima-u.ac.jp (K.M.); y-honda@asa-hosp.city.hiroshima.jp (Y.H.); 5Department of Gastroenterology, National Hospital Organization Kure Medical Center and Chugoku Cancer Center, Hiroshima 737-0023, Japan; kouno.hirotaka.pz@mail.hosp.go.jp (H.K.); kouno.hiroshi.tu@mail.hosp.go.jp (H.K.); 6Department of Gastroenterology, Chugoku Rosai Hospital, Hiroshima 737-0193, Japan; moriyat@chugokuh.johas.go.jp; 7Department of Gastroenterology, National Hospital Organization Higashihiroshima Medical Center, Hiroshima 739-0041, Japan; naeshiro.noriaki.uy@mail.hosp.go.jp; 8Department of Gastroenterology, JA Hiroshima General Hospital, Hiroshima 738-8503, Japan; andy0824hug@yahoo.co.jp (M.N.); hidehyogo@ae.auone-net.jp (H.H.); yasu-aisaka@ccv.ne.jp (Y.A.); 9Department of Gastroenterology, Hiroshima Memorial Hospital, Hiroshima 730-0802, Japan; t_azakami@yahoo.co.jp (T.A.); a-hiramatsu@kkrhiroshimakinen-hp.org (A.H.)

**Keywords:** hepatocellular carcinoma, lenvatinib, sequential therapy, molecular targeted agent, radiological response, overall survival, Response Evaluation Criteria in Solid Tumors (RECIST), modified Response Evaluation Criteria in Solid Tumors (mRECIST)

## Abstract

**Simple Summary:**

With the recent increase in the number of drug therapy options for unresectable hepatocellular carcinoma (u-HCC), the key issue has become how to prolong overall survival (OS). The aim was to evaluate the association between radiological response and OS in patients treated with lenvatinib as a first-line systemic treatment for u-HCC. Radiological response using both Response Evaluation Criteria in Solid Tumors (RECIST) and modified Response Evaluation Criteria in Solid Tumors (mRECIST) is a predictor of OS and achieving an objective response at the first evaluation is an independent prognostic factor for OS. In addition, if an objective response is obtained at the initial evaluation, continuation of treatment appears desirable because prolonged OS can be expected; but, if stable disease is obtained at the initial evaluation, one should determine whether to continue or switch to the next treatment, with careful consideration of factors related to the tumor and hepatic reserve at the initial evaluation.

**Abstract:**

The association between radiological response and overall survival (OS) was retrospectively evaluated in patients treated with lenvatinib as a first-line systemic treatment for unresectable hepatocellular carcinoma. A total of 182 patients with Child–Pugh class A liver function and an Eastern Cooperative Oncology Group performance status of zero or one were enrolled. Radiological evaluation was performed using Response Evaluation Criteria in Solid Tumors (RECIST) and modified Response Evaluation Criteria in Solid Tumors (mRECIST). Initial radiological evaluation confirmed significant stratification of OS by efficacy judgment with both RECIST and mRECIST, and that initial radiological response was an independent prognostic factor for OS on multivariate analysis. Furthermore, in patients with stable disease (SD) at initial evaluation, macrovascular invasion at the initial evaluation on RECIST and modified albumin–bilirubin grade at initial evaluation on mRECIST were independent predictors of OS on multivariate analysis. In conclusion, if objective response is obtained at the initial evaluation, continuation of treatment appears desirable because prolonged OS can be expected; but, if SD is obtained at the initial evaluation, one should determine whether to continue or switch to the next treatment, with careful consideration of factors related to the tumor and hepatic reserve at the initial evaluation.

## 1. Introduction

Hepatocellular carcinoma (HCC) is one of the major causes of cancer-related deaths worldwide, and the prognosis of unresectable HCC (u-HCC) is poor [1,2]. Until the early 2000s, the treatment of HCC focused mainly on the management of intrahepatic lesions, with remarkable advances in surgery, local ablation, and transarterial chemotherapy [3,4,5,6,7]. In 2009, sorafenib was approved as the first molecular targeted agent (MTA) for u-HCC [8], and, in 2018, lenvatinib was approved as a first-line MTA in Japan [9]. In addition, regorafenib was approved as a second-line MTA in 2017 [10], ramucirumab in 2019 [11], and cabozantinib in 2020 [12], gradually improving the prognosis of patients with u-HCC. Furthermore, with the recent development of immunotherapy, atezolizumab plus bevacizumab was approved in 2020 as the first immune combination therapy for u-HCC [13], and several other clinical trials with promising results are ongoing [14]. Recently, the effects of viral etiology on responses to immunotherapy in HCC have also come into focus [15,16].

Currently, six drug regimens are approved for the treatment of u-HCC. As these drugs can be used in multidrug sequential therapy, consideration is needed to ensure that drug switching is performed safely and effectively. Although atezolizumab plus bevacizumab is becoming established as a first-line therapy for u-HCC, the median progression-free survival (PFS) is limited to 6.9 months, and subsequent sequential drug therapy has not yet been established. Of the five currently approved MTA regimens, a particularly high response rate to lenvatinib is seen, but the problem is that it is often difficult to continue due to poor tolerability and hepatic reserve during use. To prolong overall survival (OS), the key clinical issue is how to decide whether to continue, or switch therapy based on response, tolerability, and hepatic reserve.

Therefore, in this study, the relationship between radiological response and prognosis in patients who received lenvatinib as a first-line systemic therapy was analyzed.

## 2. Materials and Methods

### 2.1. Patients

Consent to participate in this study was obtained from 250 patients who received lenvatinib for u-HCC at our hospital and affiliated institutions from April 2018 to May 2021. HCC etiology due to hepatitis C virus (HCV) or hepatitis B virus (HBV) was determined based on the presence of anti-HCV antibodies and antibodies against HBV surface antigen, respectively. The Child–Pugh classification and modified albumin–bilirubin (mALBI) grade were used to evaluate hepatic reserve. The mALBI grade was created to evaluate patients with conventional albumin–bilirubin (ALBI) grade 2 in more detail and is a four-step evaluation (ALBI score ≤ −2.60 was grade 1, −2.60 < ALBI score ≤ −2.27 was grade 2a, −2.27 < ALBI score ≤ −1.39 was grade 2b, and ALBI score > −1.39 was grade 3) [17,18]. HCC was diagnosed based on pathological or radiological features, such as early dense staining in the arterial phase, followed by a wash-out pattern in the portal/equilibrium phase on dynamic computed tomography (CT) or magnetic resonance imaging (MRI). Tumor stage was assessed using the Barcelona Clinic liver cancer (BCLC) staging system.

### 2.2. Lenvatinib Treatment Regimens

Lenvatinib was started orally at a dose of 8 mg/day for patients weighing less than 60 kg and 12 mg/day for patients weighing 60 kg or more, unless there was a specific reason not to. Adverse events were assessed using Common Terminology Criteria for Adverse Events version 5.0. In the case of drug-related adverse events, the dose of lenvatinib was reduced as necessary according to the lenvatinib dosing guidelines, and discontinued in cases of unacceptable, serious adverse events. Patients continued the therapy until death or one of the following criteria was met for the cessation of therapy: progressive disease following treatment, adverse events that required termination of treatment, deterioration of ECOG PS to 4, worsening liver function, or withdrawal of consent.

### 2.3. Assessment of Response to Lenvatinib

Radiological response assessment by dynamic CT/MRI was performed every 4–8 weeks after initiation of lenvatinib. The Response Evaluation Criteria in Solid Tumors (RECIST) version 1.1 and modified Response Evaluation Criteria in Solid Tumors (mRECIST) guidelines were used to assess treatment response, and the overall response rate (ORR) and disease control rate (DCR) were evaluated according to these guidelines. If patients obtained complete response (CR) or partial response (PR), they were defined as having achieved an objective response (OR). OS was defined as the time from initiation of lenvatinib to death from any cause. The last follow-up date was used as the censoring date for surviving patients. PFS was defined as the period from LEN initiation until the time of radiological progression by mRECIST or any cause of death.

### 2.4. Statistical Analysis

The Kaplan–Meier method, log-rank test, and Cox proportional hazards analysis were used for statistical analysis. A *p*-value less of than 0.05 was considered a statistically significant difference. All statistical analyses were performed using IBM SPSS (v. 22.0.0.0).

## 3. Results

### 3.1. Clinical Characteristics of Participating Patients

Out of 250 total patients, 182 (154 males, 28 females) who were started on first-line systemic treatment with Child–Pugh class A liver function and an Eastern Cooperative Oncology Group performance status (ECOG PS) of zero or one were included in this study. Mainly because of advanced age, five patients started at a reduced dose of 8 mg instead of the recommended 12 mg, while the others started at the recommended dose. The patients’ background characteristics are shown in Table 1. Their median age was 74 (46–90) years, and 146 patients had undergone prior non-systemic treatments, such as surgery, local ablation, and selective transarterial chemoembolization. The Child–Pugh score at the initiation of lenvatinib was 5 in 126 cases and 6 in 56 cases, and the mALBI grade was 1 in 80 cases, 2a in 51 cases, and 2b in 51 cases. 29 patients had vascular invasion, 52 patients had extrahepatic metastasis, and 16 patients had a relative tumor volume of 50% or more. The BCLC stage was B in 110 cases and C in 72 cases. The median observation period was 14.7 (0.6–38.9) months.

### 3.2. Treatment Response and Survival

The median OS and PFS of the 182 patients included in the study were 20.2 months and 8.1 months, respectively (Figure 1). Table 2 shows the radiological response at the first, second, and best times by RECIST and mRECIST evaluations. On the initial radiological response evaluation, 41 patients (23.7%) had OR, 102 (59.0%) had stable disease (SD), and 30 (17.3%) had progressive disease (PD) on RECIST evaluation (ORR 23.7%, DCR 82.7%), and 80 patients (47.6%) had OR, 60 (35.7%) had SD and 28 (16.7%) had PD on mRECIST evaluation (ORR 47.6%, DCR 83.3%). Similarly, good ORR and DCR were confirmed on both RECIST and mRECIST evaluations at the second and best response evaluations.

### 3.3. OS for Each Initial Radiological Response and Prognostic Factors for OS

The median OS by initial radiological response on RECIST was not reached in the OR group, but was 25.4 months in the SD group and 9.1 months in the PD group, while the median OS by initial radiological response on mRECIST was 32.1 months in the OR group, 19.3 months in the SD group, and 9.1 months in the PD group. Both RECIST and mRECIST evaluations showed significant OS stratification by response (RECIST: *p* < 0.005, mRECIST: *p* < 0.005). Similarly, the best and second radiological evaluations also showed significant stratification of OS for each response (Figure 2).

Next, the prognostic factors for OS were examined in patients treated with lenvatinib by univariate and multivariate analyses (Table 3). To avoid confounding factors, analyses were performed separately for RECIST and mRECIST. On multivariate analysis with factors including RECIST, etiology (hazard ratio, 0.605; 95% confidence interval, 0.380–0.962; *p* = 0.034), mALBI at initiation (hazard ratio, 0.409; 95% confidence interval, 0.249–0.674; *p* < 0.005), serum alpha-fetoprotein (AFP) level at initiation (hazard ratio, 0.409; 95% confidence interval, 0.251–0.667; *p* < 0.005), and initial radiological response on RECIST (hazard ratio, 0.369; 95% confidence interval, 0.197–0.691; *p* < 0.005) were independent prognostic factors for OS. On multivariate analysis of factors including mRECIST, the following were identified as independent prognostic factors for OS: mALBI at initiation (hazard ratio, 0.451; 95% confidence interval, 0.277–0.734; *p* < 0.005), serum AFP level at initiation (hazard ratio, 0.359; 95% confidence interval, 0.221–0.583; *p* < 0.005), and initial radiological response on mRECIST (hazard ratio, 0.378; 95% confidence interval, 0.234–0.611; *p* < 0.005). On both RECIST and mRECIST evaluations, good mALBI at initiation (1–2a), low AFP at initiation, and obtaining OR at the initial radiological response evaluation were extracted as independent prognostic factors for OS in lenvatinib.

### 3.4. Prognostic Factors for OS in Patients with SD at the Initial Radiological Response Evaluation

Since it has been shown that obtaining OR at the initial radiological response evaluation contributes to longer OS, patients with SD at the initial radiological response evaluation were examined next. On univariate and multivariate analyses, the independent prognostic factors for OS from the initial response evaluation were examined separately for RECIST and mRECIST (Table 4). In patients with SD at the initial response evaluation on RECIST, macrovascular invasion (MVI) at the time of initial evaluation (hazard ratio, 0.347; 95% confidence interval, 0.143–0.843; *p* = 0.019) was the independent prognostic factor for OS from the initial evaluation on multivariate analysis. The median OS from the initial evaluation on RECIST was 28.8 months in the group without MVI at the initial evaluation and 9.7 months in the group with MVI (Figure 3). In patients with SD at the initial response evaluation on mRECIST, mALBI at the time of initial evaluation (hazard ratio, 0.381; 95% confidence interval, 0.156–0.932; *p* = 0.035) was an independent prognostic factor for OS from the initial evaluation on multivariate analysis. The median OS from the initial evaluation on mRECIST was 24.4 months in the group with mALBI of 1–2a at the initial evaluation and 10.6 months in the group with mALBI of 2b (Figure 3). On both RECIST and mRECIST evaluations, the second radiological response was not a prognostic factor on univariate analysis.

## 4. Discussion

In this study, the effect of radiological response on prognosis in patients with u-HCC who received lenvatinib as a first-line systemic treatment was investigated. The results showed that initial radiological evaluations based on either RECIST or mRECIST were each stratified with respect to OS, and that the initial radiological response was an independent prognostic factor for OS. In addition, stratification of OS by response was confirmed not only for the initial response, but also for the second and best responses.

Next, univariate and multivariate analyses of factors contributing to OS from the initial radiological evaluation in patients with SD at the initial response evaluation were performed. It was found that MVI at the time of the initial response evaluation was an independent prognostic factor for OS from the initial radiological evaluation on RECIST, and mALBI grade at the time of the initial response evaluation was an independent prognostic factor for OS from the initial radiological evaluation on mRECIST. On the other hand, the second radiological response was not a prognostic factor for OS from the initial radiological response based on either RECIST or mRECIST evaluations. These results suggest that, in patients with SD at the initial radiological evaluation, the decision to continue the current treatment or switch to the next treatment should be made with careful consideration of factors related to the tumor and hepatic reserve at the time of the initial radiological evaluation, rather than simply continuing and waiting for the second response evaluation.

Several predictors for the efficacy of lenvatinib (4-week relative dose intensity [19], AFP [20,21], ALBI grade [20,21,22], neutrophil-to-lymphocyte ratio [23], and occurrence of hypothyroidism [24]) have been reported previously. On radiological evaluation, obtaining OR by mRECIST evaluation has been reported to be an independent predictor of OS with other MTAs [25,26,27]. Kaneko et al. reported that early evaluation by RECIST 1.1 was useful for prognostic stratification in lenvatinib [28], and Kudo et al. reported that objective response by mRECIST evaluation was associated with OS in a multivariate analysis of responders to lenvatinib in the REFLECT trial [29]. Hiraoka et al. also reported that, when ECOG PS and hepatic reserve function permit, continuing lenvatinib beyond PD, especially in u-HCC patients who showed a hand–foot skin reaction during lenvatinib treatment, might be a good therapeutic option [30]. One of the characteristics of lenvatinib is that it has shorter time to response compared to other drugs, especially atezolizumab plus bevacizumab, which is becoming established as a first-line treatment for u-HCC. In order to evaluate the efficacy of lenvatinib effectively, it is very important to confirm the response early in the course of treatment, when the relative dose intensity can be relatively maintained and is less susceptible to intolerance and loss of hepatic reserve.

If the radiological evaluation shows OR, it is desirable to continue the treatment, as long as it is well tolerated because good OS prolongation can be expected. Conversely, if the radiological evaluation shows PD, it would be desirable to consider switching to the next treatment. However, if the radiological response shows SD, there is no clear consensus on whether to continue or to switch to the next treatment. In the REFLECT trial, lenvatinib showed good ORR, but PFS was limited to about 7.4 months. The major causes of this limitation are decreased tolerability, decreased hepatic reserve, and acquisition of tolerance. In order to prolong OS, which is the main goal in the treatment of HCC, we should be more careful about making decisions when the radiological response is SD. This study is the first to examine the direction of treatment using multivariate analysis of factors contributing to OS when the radiological response is SD, after confirming the stratification of OS by radiological response.

In this study, there was a clear difference in OS of 25.4 months for RECIST and 19.3 months for mRECIST in the group with SD on initial radiological evaluation, compared to the group with OR and PD. The mRECIST evaluation is a valid evaluation method for u-HCC, especially with an MTA with angiogenesis inhibition, and it has been used along with the RECIST evaluation for radiological evaluation of HCC. The reason for this may be that the mRECIST evaluation more sensitively shows antitumor effects, as reflected in the loss of staining compared to the RECIST evaluation. Kuzuya et al. indicated that radiological antitumor response by mRECIST (i.e., disappearance of arterial tumor enhancement) may not necessarily reflect tumor necrosis, especially soon after initiation of lenvatinib [31]; thus, careful judgment is needed when using mRECIST. Nevertheless, the SD evaluation by mRECIST (i.e., no disappearance of arterial tumor enhancement) may indicate that even the anti-tumor effect, which is the most important feature of lenvatinib, was not achieved, so more attention may be required in treatment planning than the SD evaluation by RECIST.

In recent years, with the increase of new drug therapies in u-HCC, the treatment paradigm has changed, and the selection, timing, and sequence of appropriate therapies have become major issues. The IMbrave150 trial demonstrated that atezolizumab plus bevacizumab combination therapy significantly improved median OS (not reached vs. 13.2 months; hazard ratio 0.58, *p* < 0.001), median PFS per RECIST version 1.1 (6.8 vs. 4.3 months; HR 0.59; *p* < 0.001), and ORR per RECIST version 1.1 (27 vs. 12%; *p* < 0.001) compared to sorafenib, and also maintained quality of life [16]. As a result, atezolizumab plus bevacizumab is expected to be a useful systemic therapy for u-HCC and is positioned as a first-line treatment, but the median PFS is limited to 6.9 months, as reported by the updated analysis of IMbrave150. Due to the favorable safety profile and quality of life of atezolizumab plus bevacizumab, it is expected that many patients will be able to maintain hepatic reserve after progression and move on to the next treatment, but the optimal sequence of atezolizumab plus bevacizumab after progression has not yet been established. Yoo et al. reported that second-line treatment with sorafenib and lenvatinib after progression on atezolizumab plus bevacizumab was as effective as these MTAs in the pivotal phase 3 trials [32]. The efficacy of sequential multidrug therapy in u-HCC has been reported in several studies and further investigation is needed, including atezolizumab plus bevacizumab [33,34]. Alsina et al. reported that, in a post hoc analysis of patients entered into the REFLECT trial, first-line lenvatinib followed by subsequent systemic therapy led to a longer OS and may provide greater survival benefit in patients who achieved OR to lenvatinib [35]. How to effectively manage lenvatinib, which has a particularly high response rate among MTAs, is considered to be a very important issue. A large-scale prospective observational study is currently being conducted in Japan to collect real-world data on sequential systemic drug therapy for HCC and is expected to establish effective treatment regimens through further data collection [36].

It is still unclear which systemic therapy to prefer as the next therapy after lenvatinib. One of the problems with lenvatinib is that it often requires dose reduction or withdrawal due to adverse events or loss of hepatic reserve, regardless of response. It is very important to determine the appropriate timing for switching to the next therapy while maintaining hepatic reserve without unreasonably continuing lenvatinib. In this study, responders showed clearly better OS as in previous reports, and we also observed a clear stratification of OS in patients with SD and PD. On the other hand, poor prognosis was observed in patients with SD who had poor hepatic reserve (mALBI grade 2b) or MVI at the initial evaluation. These results suggest that it may be possible to stratify patients with SD, especially by focusing on hepatic reserve and MVI at the initial evaluation, so that we can consider the appropriate timing of switching from lenvatinib for OS prolongation earlier. With the current availability of multiple systemic therapies for HCC, it may be necessary to consider switching to other MTAs even if SD is achieved with lenvatinib. In the near future, it is expected that further systemic treatment options will become available with the results of ongoing clinical trials focusing on immunotherapy. It is known that a certain number of patients are refractory to immunotherapy in terms of tumor microenvironment, but it has been suggested that the use of MTAs, including lenvatinib, in combination with immunotherapy or as the next treatment may lead to response. Therefore, the management of MTAs (especially lenvatinib) and the establishment of multidrug sequential therapy will continue to be important clinical issues. In the current paradigm shift in the systemic treatment of u-HCC, we need to continue to accumulate clinical cases to establish evidence.

This study had several limitations as a retrospective study with a small sample size, an insufficient observation period, a reduced starting dose in some patients, and time bias due to the timing of radiological evaluations (every 4–8 weeks). In addition, a prospective study with a longer observation period and a larger number of patients is needed to draw a more definitive conclusion on whether switching from lenvatinib actually leads to prolonged OS. Nevertheless, we believe that the results of this study will have an important impact on the decision-making process during treatment with lenvatinib.

## 5. Conclusions

Radiological response on both RECIST and mRECIST evaluations stratifies OS, and achieving objective response at the first evaluation is an independent prognostic factor for OS. In addition, if the initial evaluation shows SD, it may be important to consider factors related to the tumor and hepatic reserve when determining treatment strategy.

## Figures and Tables

**Figure 1 cancers-14-00320-f001:**
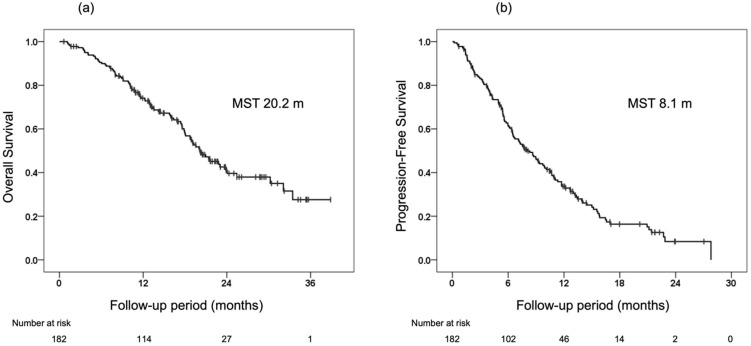
Overall survival (OS) and progression-free survival (PFS) from the initiation of lenvatinib in the 182 patients included in this study. (**a**) OS from the initiation of lenvatinib. (**b**) PFS from the initiation of lenvatinib.

**Figure 2 cancers-14-00320-f002:**
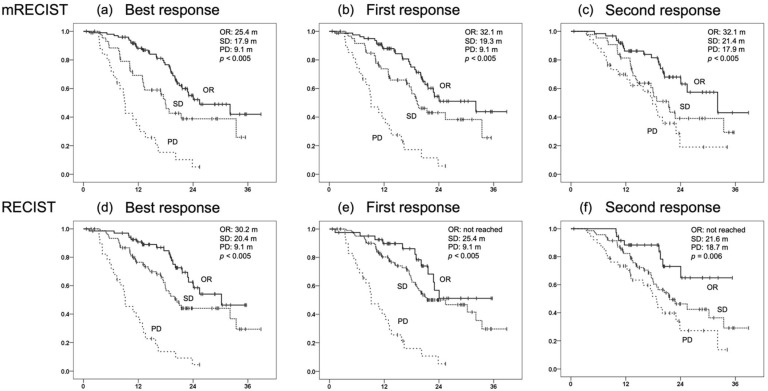
Comparison of overall survival (OS) by response at the first, second, and best responses evaluated by Response Evaluation Criteria in Solid Tumors (RECIST) and modified Response Evaluation Criteria in Solid Tumors (mRECIST). (**a**) OS at the best response evaluated by mRECIST (OR 25.4 months, SD 17.9 months, PD 9.1 months, *p* < 0.005). (**b**) OS at the first response evaluated by mRECIST (OR 32.1 months, SD 19.3 months, PD 9.1 months, *p* < 0.005). (**c**) OS at the second response evaluated by mRECIST (OR 32.1 months, SD 21.4 months, PD 17.9 months, *p* < 0.005). (**d**) OS at the best response evaluated by RECIST (OR 30.2 months, SD 20.4 months, PD 9.1 months, *p* < 0.005). (**e**) OS at the first response evaluated by RECIST (OR not reached, SD 25.4 months, PD 9.1 months, *p* < 0.005). (**f**) OS at the second response evaluated by RECIST (OR not reached, SD 21.6 months, PD 18.7 months, *p* = 0.006).

**Figure 3 cancers-14-00320-f003:**
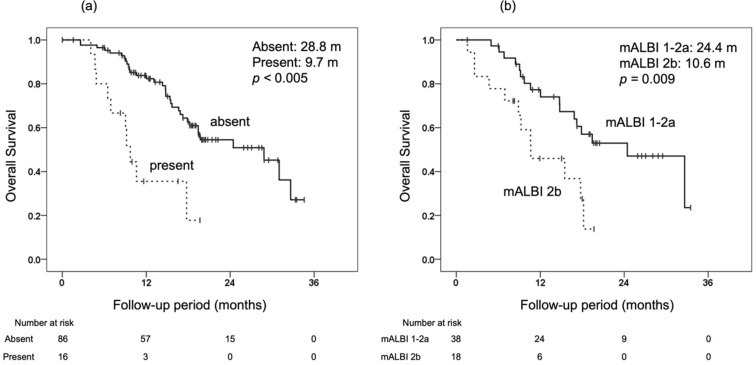
Comparison of overall survival (OS) from the initial objective evaluation of lenvatinib in patients with SD at the initial radiological response evaluation. (**a**) OS from the initial objective evaluation by RECIST with or without macroscopic vascular invasion (absent 28.8 months, present 9.7 months, *p* < 0.005). (**b**) OS from the initial objective evaluation by mRECIST by mALBI grade (mALBI 1–2a 24.4 months, mALBI 2b 10.6 months, *p* = 0.009).

**Table 1 cancers-14-00320-t001:** Clinical characteristics at the initiation of lenvatinib (*n* = 182).

Characteristic	Median (Range) or Patients, *n*
Age, range, y	74 (46–90)
Sex (male/female), *n*	154/28
Weight (<60/>60 kg), *n*	84/98
Performance status (0/1), *n*	166/16
Etiology (HBV/HCV/HBV + HCV/NBNC), *n*	22/62/1/97
History of non-systemic treatment (with/without), *n*	146/36
Total bilirubin, range, mg/dL	0.8 (0.3–2.1)
Albumin, range, g/dL	3.9 (2.9–4.9)
Prothrombin activity, range, %	90 (59–131)
Child–Pugh score (5/6), *n*	126/56
mALBI grade (1/2a/2b), *n*	80/51/51
Size of main tumor, range, mm	24.0 (0.0–190.0)
Relative tumor volume (<50/≥50%), *n*	166/16
Macroscopic vascular invasion (absent/present), *n*	153/29
Extrahepatic metastasis (absent/present), *n*	130/52
BCLC stage (B/C), *n*	110/72
Serum AFP value, range, ng/mL	20.2 (0.5–236900.0)
Serum DCP value, range, mAU/mL	174.0 (13.0–1083990.0)
Observation period, range, months	14.7 (0.6–38.9)

HBV, hepatitis B virus infection; HCV, hepatitis C virus infection; NBNC, non-B-non-C viral hepatitis; mALBI, modified albumin–bilirubin; BCLC, Barcelona Clinic liver cancer; AFP, alpha-fetoprotein; DCP, des-γ-carboxy prothrombin.

**Table 2 cancers-14-00320-t002:** Radiological responses to lenvatinib.

Response	RECIST % (*n*)			mRECIST % (*n*)		
	Best	1st	2nd	Best	1st	2nd
CR	4.0 (7)	2.3 (4)	2.1 (3)	17.1 (29)	7.7 (13)	12.9 (18)
PR	35.6 (62)	21.4 (37)	23.4 (34)	42.4 (72)	39.9 (67)	32.1 (45)
SD	44.3 (77)	59.0 (102)	48.3 (70)	25.9 (44)	35.7 (60)	30.7 (43)
PD	16.1 (28)	17.3 (30)	26.2 (38)	14.7 (25)	16.7 (28)	24.3 (34)
ORR	39.7 (69)	23.7 (41)	25.5 (37)	59.4 (101)	47.6 (80)	45.0 (63)
DCR	83.9 (146)	82.7 (143)	73.8 (107)	85.3 (145)	83.3 (140)	75.7 (106)

RECIST, Response Evaluation Criteria in Solid Tumors; mRECIST, modified Response Evaluation Criteria in Solid Tumors; CR, complete response; PR, partial response; SD, stable disease; PD, progressive disease; ORR, overall response rate; DCR, disease control rate.

**Table 3 cancers-14-00320-t003:** Univariate and multivariate analyses of prognostic factors for overall survival.

Factors	Univariate *p*-Value	Multivariate		
		HR	95% CI	*p*-Value
Age (<74 vs. ≥74 years)	0.302			
Sex (male vs. female)	0.279			
Etiology (NBNC vs. viral)	0.010	0.605	0.380–0.962	0.034
History of non-systemic treatment (with vs. without)	0.981			
mALBI grade (1/2a vs. 2b)	<0.005	0.409	0.249–0.674	<0.005
Macroscopic vascular invasion (absent vs. present)	<0.005	0.838	0.320–1.129	0.113
Extrahepatic metastasis (absent vs. present)	0.010	0.601	0.456–1.199	0.221
Relative tumor volume (<50% vs. ≥50%)	<0.005	0.740	0.377–1.866	0.666
Serum AFP value (<400 vs. ≥400), ng/mL	<0.005	0.409	0.251–0.667	<0.005
Serum DCP value (<174 vs. ≥174), ng/mL	0.133			
Initial objective response by RECIST (OR vs. non-OR)	0.007	0.369	0.197–0.691	<0.005
Age (<74 vs. ≥74 years)	0.302			
Sex (male vs. female)	0.279			
Etiology (NBNC vs. viral)	0.010	0.662	0.416–1.055	0.083
History of non-systemic treatment (with vs. without)	0.981			
mALBI grade (1/2a vs. 2b)	<0.005	0.451	0.277–0.734	<0.005
Macroscopic vascular invasion (absent vs. present)	<0.005	0.861	0.437–1.697	0.666
Extrahepatic metastasis (absent vs. present)	0.010	0.786	0.484–1.278	0.332
Relative tumor volume (<50% vs. ≥50%)	<0.005	0.488	0.215–1.111	0.087
Serum AFP value (<400 vs. ≥400), ng/mL	<0.005	0.359	0.221–0.583	<0.005
Serum DCP value (<174 vs. ≥174), ng/mL	0.133			
Initial objective response by mRECIST (OR vs. non-OR)	<0.005	0.378	0.234–0.611	<0.005

NBNC, non-B-non-C viral hepatitis; mALBI, modified albumin–bilirubin; AFP, alpha-fetoprotein; DCP, des-γ-carboxy prothrombin; RECIST, Response Evaluation Criteria in Solid Tumors; mRECIST, modified Response Evaluation Criteria in Solid Tumors; OR, objective response.

**Table 4 cancers-14-00320-t004:** Univariate and multivariate analyses of prognostic factors for overall survival from initial objective evaluation in patients with SD at the initial radiological response evaluation.

Factors	Univariate *p*-Value	Multivariate		
		HR	95% CI	*p*-Value
Age (<74 vs. ≥74 years)	0.444			
Sex (female vs. male)	0.072			
Etiology (NBNC vs. viral)	0.016	0.584	0.303–1.124	0.107
History of non-systemic treatment (with vs. without)	0.555			
mALBI grade at initial objective evaluation (1/2a vs. 2b)	0.031	0.743	0.352–1.567	0.435
Decrease in AFP value up to initial objective evaluation (yes vs. no)	0.482			
Decrease in DCP value up to initial objective evaluation (yes vs. no)	0.574			
Relative dose intensity up to initial objective evaluation (<0.8 vs. ≥0.8)	0.540			
Macroscopic vascular invasion at initial objective evaluation (absent vs. present)	<0.005	0.347	0.143–0.843	0.019
Extrahepatic metastasis at initial objective evaluation (absent vs. present)	0.169			
Relative tumor volume at initial objective evaluation (<50% vs. ≥50%)	0.005	0.464	0.158–1.361	0.162
Second objective response by RECIST (OR vs. non-OR)	0.225			
Age (<74 vs. ≥74 years)	0.946			
Sex (female vs. male)	0.542			
Etiology (NBNC vs. viral)	0.052			
History of non-systemic treatment (with vs. without)	0.911			
mALBI grade at initial objective evaluation (1/2a vs. 2b)	0.009	0.381	0.156–0.932	0.035
Decrease in AFP value up to initial objective evaluation (yes vs. no)	0.323			
Decrease in DCP value up to initial objective evaluation (yes vs. no)	0.848			
Relative dose intensity up to initial objective evaluation (<0.8 vs. ≥0.8)	0.302			
Macroscopic vascular invasion at initial objective evaluation (absent vs. present)	0.013	0.671	0.247–1.824	0.435
Extrahepatic metastasis at initial objective evaluation (absent vs. present)	0.212			
Relative tumor volume at initial objective evaluation (<50% vs. ≥50%)	<0.005	0.216	0.042–1.114	0.067
Second objective response by mRECIST (OR vs. non-OR)	0.443			

SD, stable disease; NBNC, non-B-non-C viral hepatitis; mALBI, modified albumin–bilirubin; AFP, alpha-fetoprotein; DCP, des-γ-carboxy prothrombin; RECIST, Response Evaluation Criteria in Solid Tumors; mRECIST, modified Response Evaluation Criteria in Solid Tumors; OR, objective response.

## Data Availability

The data that support the findings of this study are available from the corresponding author upon reasonable request.

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
