# Peer review of "Analysis of Survival and Response to Lenvatinib in Unresectable Hepatocellular Carcinoma"

_cancers, 2022, doi:10.3390/cancers14020320_

Round 1
Reviewer 1 Report
Dear Editor, thank you so much for inviting me to revise this manuscript about HCC.
This study addresses a current topic.
The manuscript is quite well written and organized. English could be improved.
Figures and tables are comprehensive and clear.
The introduction explains in a clear and coherent manner the background of this study.
We suggest the following modifications:
- Introduction section: although the authors correctly included important papers in this setting, we believe a couple of studies should be cited within the introduction (PMID: 34167433; PMID: 34659220 ), only for a matter of consistency. We think it might be useful to introduce the topic of this interesting study.
- Methods and Statistical Analysis: nothing to add.
- Discussion section: Very interesting and timely discussion. Of note, the authors should expand the Discussion section, including a more personal perspective to reflect on. For example, they could answer the following questions – in order to facilitate the understanding of this complex topic to readers: what potential does this study hold? What are the knowledge gaps and how do researchers tackle them? How do you see this area unfolding in the next 5 years? We think it would be extremely interesting for the readers.
However, we think the authors should be acknowledged for their work. In fact, they correctly addressed an important topic in HCC, the methods sound good and their discussion is well balanced.
One additional little flaw: the authors could better explain the limitations of their work, in the last part of the Discussion.
We believe this article is suitable for publication in the journal although some revisions are needed. The main strengths of this paper are that it addresses an interesting and very timely question and provides a clear answer, with some limitations.
We suggest a linguistic revision and the addition of some references for a matter of consistency. Moreover, the authors should better clarify some points.
Reviewer 2 Report
Thank you so much for the opportunity of the review of the manuscript entitled “Analysis of Survival and Response to Lenvatinib in Unresectable Hepatocellular Carcinoma”.
A total of 182 patients who received lenvatinib as a first-line therapy and Child-Pugh class A liver function and Eastern Cooperative Oncology Group performance status 0 or 1 were enrolled in this study. The relationship between radiological response and prognosis was analyzed. Authors concluded if objective response was obtained at the initial evaluation, continuation of treatment appears desirable, because prolonged OS could be expected, but if stable disease was obtained at the initial evaluation, one should determine whether to continue or switch to the next treatment with careful consideration of factors related to the tumor and hepatic reserve at the initial evaluation.
I agree that effective management of lenvatinib is very important for achieving a longer OS.
I have several comments.
A total of 182 patients with Child-Pugh class A liver function and Eastern Cooperative Oncology Group performance status 0 or 1 were enrolled in this study. Was lenvatinib used as the first-line systemic therapy? In some cases, previous therapy such as RFA and/or TACE might be present. Or, was lenvatinib used as the initial therapy (no previous therapy was present) of HCC? If lenvatinib was used as the first-line systemic therapy, do previous treatments affect prognostic factors for OS?
Did all patients start with the recommended dose?
Authors described that dynamic CT/MRI was performed every 4-8 weeks after initiation of lenvatinib. Some patients underwent dynamic CT/MRI at 4 weeks as the initial evaluation and 8 weeks as the second evaluation after initiation of lenvatinib. In contrast, some patients underwent dynamic CT/MRI at 8 weeks as the initial evaluation and 16 weeks as the second evaluation after initiation of lenvatinib. How to deal with these evaluation points.
As authors described that, in patients with stable disease (SD) at initial evaluation, macrovascular invasion at the initial evaluation on RECIST, and modified albumin-bilirubin grade at initial evaluation on mRECIST were independent predictors of OS on multivariate analysis. To avoid confusion, I have questions. Was macrovascular invasion (MVI) observed at initial evaluation on RECIST (some cases had no MVI at the start of lenvatinib)? Was mALBI grade evaluated again at initial evaluation on mRECIST (some cases were different from mALBI at the start of lenvatinib)?
Author described that, If SD is obtained at the initial evaluation, one should determine whether to continue or switch to the next treatment with careful consideration of factors related to the tumor and hepatic reserve at the initial evaluation.
There may not be correct answers, if the hepatic reserve is not sufficient (mALBI grade is 2b), how should we specifically change the treatment?
Round 2
Reviewer 1 Report
The authors modified the manuscript according to our suggestions.
We recommend Acceptance.